# A Statistical Approach for Functional Reach-to-Grasp Segmentation Using a Single Inertial Measurement Unit

**DOI:** 10.3390/s24186119

**Published:** 2024-09-22

**Authors:** Gregorio Dotti, Marco Caruso, Daniele Fortunato, Marco Knaflitz, Andrea Cereatti, Marco Ghislieri

**Affiliations:** Polito^BIO^Med Lab, Department of Electronics and Telecommunications, Politecnico di Torino, 10129 Turin, Italy; marco.caruso@polito.it (M.C.); daniele.fortunato@polito.it (D.F.); marco.knaflitz@polito.it (M.K.); andrea.cereatti@polito.it (A.C.); marco.ghislieri@polito.it (M.G.)

**Keywords:** activity of daily living, functional assessment, IMU, movement segmentation, telerehabilitation, upper limb

## Abstract

The aim of this contribution is to present a segmentation method for the identification of voluntary movements from inertial data acquired through a single inertial measurement unit placed on the subject’s wrist. Inertial data were recorded from 25 healthy subjects while performing 75 consecutive reach-to-grasp movements. The approach herein presented, called DynAMoS, is based on an adaptive thresholding step on the angular velocity norm, followed by a statistics-based post-processing on the movement duration distribution. Post-processing aims at reducing the number of erroneous transitions in the movement segmentation. We assessed the segmentation quality of this method using a stereophotogrammetric system as the gold standard. Two popular methods already presented in the literature were compared to DynAMoS in terms of the number of movements identified, onset and offset mean absolute errors, and movement duration. Moreover, we analyzed the sub-phase durations of the drinking movement to further characterize the task. The results show that the proposed method performs significantly better than the two state-of-the-art approaches (i.e., percentage of erroneous movements = 3%; onset and offset mean absolute error < 0.08 s), suggesting that DynAMoS could make more effective home monitoring applications for assessing the motion improvements of patients following domicile rehabilitation protocols.

## 1. Introduction

Activities of daily living (ADLs) are fundamental for independent living and, in this regard, the functionality of the upper limbs is crucial for a good quality of life [1,2]. Unfortunately, 28% of the population over 50 years of age and 50% of the population over 80 years of age are affected by movement disorders [3].

To define appropriate interventions for motor disorders management, an accurate clinical assessment sets the basis for designing a successful motor rehabilitation program and for testing its effectiveness. Clinical assessment is usually performed using scales that grade movement disorders based on the clinician’s evaluation of a specific task. However, in recent decades, many motion analysis systems have been proposed and used for reducing the subjectivity in patient clinical evaluation, enhancing the effectiveness of rehabilitation outcome evaluation, especially for the upper limbs [4,5,6,7,8]. In particular, inertial measurement units (IMUs) have been widely used for the assessment and rehabilitation of movement disorders of the upper limbs [9]. IMUs measure the acceleration and angular velocity of the body segment they are fixed to, allowing for the quantitative analysis of patient movements [9]. The use of IMUs arose thanks to their ease of use, portability, and low cost. For example, using IMUs allows clinicians to tailor rehabilitation protocols to the patient’s needs [10] and, in the context of therapy delivery systems, allows patients to decide when and where to carry out therapeutic sessions [11]. However, partly due to methodological issues, the use of inertial technology for measuring the quality of movements during functional tasks in a clinical environment is still not fully exploited [12]. Therefore, there is a need to develop and validate new methods that increase the reliability and validity of IMU-derived evaluation metrics in clinics [12].

Among all the proposed parameters, execution time is one of the most commonly used metrics for the assessment of patient functionality in a clinical context. Hence, it is necessary to precisely identify voluntary movements. To this end, several methods based on inertial data have been presented in the literature. The most straightforward approach is to define a threshold that discriminates between voluntary and involuntary movements. For example, Schwarz et al. [13] identified voluntary movements by applying a fixed threshold to the angular velocity norm. Voluntary movements were identified in correspondence to the time instants above this threshold. Carpinella et al. [14] proposed a similar approach, setting an adaptive threshold at 25% of the maximum of the angular velocity norm during the movement. Setting the threshold value equal to a percentage of the maximum recorded value guarantees that the threshold is more suited to the characteristics of the subject under analysis, reducing the influence of inter-subject variability on the segmentation results. Moreover, Hughes et al. [15] identified voluntary movement onset and offset by applying a kinematic criterion on the linear velocity derived from the IMU accelerometer. In detail, the onset was determined as the first instance in the time series wherein the resultant velocity exceeded 1.5% of the first velocity peak, and the offset was determined as when the velocity dropped under 1.5% of the velocity peak. The method proposed by Hughes et al. presents several challenges. First, the reconstruction of the linear velocity relies on numerically integrating the linear acceleration after removing the gravitational bias. This process requires accurately estimating the sensor orientation using a sensor fusion filter. Even with fine-tuning of the filter parameters, residual errors persist, leading to propagation errors in the numerical integration process [16]. Additionally, the initial conditions of orientation and velocity are critical factors that significantly impact the accuracy of the results. Aoki et al. [17] proposed a segmentation method using a k-NN classifier based on data recorded from three IMUs. Specifically, the angular velocity norm is computed for each sensor, and then the data are windowed. Each observation window is classified by the k-NN into one of two states: “static” or “moving”. Following classification, post-processing is applied to remove short transitions between states. Cui et al. [18] proposed a segmentation method based on the analysis of relative rotation obtained from the IMU. The method applies a set of rules to the relative rotation of each of several sensors over a temporal observation window to differentiate between the moving and static states. Repnik et al. [19] presented a different segmentation method, based on the biomechanical model reconstruction of the upper limb from multiple IMUs mounted on the patient chest, arm, and forearm. Apart from the approaches used by Schwarz et al. [13] and Cui et al. [18], all the other methods cannot be used in real-time applications since they need to extract information from the whole IMU recordings before performing movement segmentation.

Among the aforementioned methods, the most used approaches for voluntary movement segmentation are those presented by Schwarz et al. [13] and Carpinella et al. [14], due to their simplicity and scalability. Nevertheless, both methods are subject to limitations. The segmentation method proposed by Schwarz et al. [13] does not present a technical validation and a threshold optimization process. Moreover, the application of a fixed arbitrary threshold may strongly reduce the adaptability of the method to different movements and subject characteristics. These limitations have been partially solved by Carpinella et al. [14], who performed a technical validation (testing several adaptive thresholds) and employed an adaptive threshold to identify voluntary movement from the angular velocity norm. However, the definition of the adaptive threshold can have a significant impact on the final segmentation results. The choice of a high value (e.g., 25% of the maximum angular velocity norm) could lead to the exclusion of parts of movements or even whole movements. On the other hand, selecting a value that is too low may lead to the inclusion of involuntary movements or background noise. This is especially true when analyzing movements that are composed of sub-phases executed at very different intensity levels. Moreover, the reliability of the segmentation results may be compromised by signal fluctuations around the threshold value, resulting in fast erroneous transitions.

To overcome the limitations of the approaches proposed by Schwarz et al. [13] and Carpinella et al. [14], we developed a new method and performed a technical validation using a StereoPhotogrammetric (SP) system as a gold standard. The newly proposed segmentation method, DynAMoS (Dynamic Adaptive Movement Segmentation), enhances the state-of-the-art adaptive thresholding approach with statistics-based post-processing aimed at reducing erroneous segmentation by applying statistical considerations to the movement duration histograms. In this paper, after the evaluation of the effectiveness of the proposed method against the gold standard and state-of-the-art approaches (i.e., Schwarz et al. [13] and Carpinella et al. [14]), segmentation results are used for the characterization of a reach-to-grasp movement. With the aim of supporting the adoption and standardization of IMU-derived parameters in clinics, ISB recommendations on the use of wearable measurement technology were followed [20], and we made the DynAMoS Matlab algorithm freely available, its detailed documentation, and a sample dataset on the BIOLAB GitHub repository (https://github.com/Biolab-PoliTO/DynAMoS (accessed on 9 August 2024)).

## 2. Materials and Methods

### 2.1. Participants

Twenty-five healthy subjects (12 females and 13 males; age: 22.5 ± 2.1 years; 6 left-handed and 19 right-handed) participated in the study. To participate, volunteers were required to have no history of physical or neurological pathologies that might interfere with their ability to perform the task. The subject height and weight were recorded by self-report. The dominant forearm length was measured with a flexible measuring tape with the forearm facing downward, measuring from the lateral epicondyle to the ulnar styloid process.

This study was approved by the Ethics Committee of Politecnico di Torino (Protocol N. 24766/2022, approved on 19 July 2022). Written informed consent was obtained from each participant before the experimental sessions and all the acquisitions were performed following the Declaration of Helsinki.

### 2.2. Acquisition System

Recordings were carried out at the Motion Analysis Laboratory of Polito^BIO^Med Lab, Politecnico di Torino (Turin, Italy). Inertial data were recorded using an IMU-based wearable device designed and developed at BIOLAB, Politecnico di Torino (Turin, Italy). This device incorporates an IMU featuring a three-axial accelerometer and gyroscope (LSM9DS1, STMicroelectronics, Geneva, Switzerland; fullscale: ±16 g, and ±2000 dps), a Bluetooth Low Energy module, a floating-point microcontroller (SAME70, Microchip, Chandler, Arizona, USA) to easily install and run custom algorithms onboard, a micro-SD card to store raw and processed data, and a rechargeable battery with a capacity of 1 Ah [21]. The device was calibrated following the approach described by Stančin et al. [22]. All the IMU recordings were acquired at a sampling frequency of 100 Hz.

A stereophotogrammetric system consisting of twelve infrared cameras (Vicon T20, Vicon Motion Systems, Yarnton, Oxfordshire; sampling frequency: 100 Hz) was used to reconstruct the trajectories of 4 photo-reflective markers (diameter: 9.5 mm) attached to the IMU. The IMU was secured to the wrist of the subject using double-sided adhesive tape [20] with its short edge roughly aligned with the wrist’s flexion–extension axis.

Three RGB cameras integrated with the SP system were used to record the acquisitions (sampling frequency: 50 Hz). Video recordings were anonymized by blurring the subject’s face.

Figure 1a shows a schematic representation of the acquisition system.

The IMU and SP signals were then imported into MATLAB release r2023b (The MathWorks Inc., Natick, MA, USA) to be processed offline through custom routines.

### 2.3. Experimental Protocol

Volunteers were seated at a table (distance between the tabletop and the seat: 30 cm; table height: 70 cm) and were asked to perform a drinking task using their dominant upper limb. The bottle was positioned in front of the subject’s sternum at a distance from the table edge equal to 1.5 times the forearm length. The drinking task consisted of reaching for and grasping the bottle (Phase I), lifting the bottle simulating drinking (Phase II), placing the bottle back on the table (Phase III), and returning to the resting position (Phase IV). Figure 1b schematically represents the experimental protocol with the indication of the three task sub-phases. The investigated task represents a typical ADL and is part of the Frenchay Arm Test, which is commonly used to evaluate upper limb function [23]. The position of the bottle on the table and the wrist’s resting position were marked using adhesive tape, to ensure the repeatability of the movement.

Before the acquisition session, a 1 min static acquisition was recorded to compensate for IMU biases [24]. At the start of each trial, subjects were requested to perform a 30 s static acquisition, after which they were instructed to raise the instrumented arm (dominant side) and perform three rapid rotations along the forearm longitudinal axis before returning to the resting position for 30 s. This movement was necessary to synchronize the SP system and the wrist-worn IMU. Subsequently, subjects were required to perform 25 repetitions of the drinking task. To align with rehabilitation practice, the drinking task sub-phases were executed at a self-selected pace following the verbal instructions given by the investigator. Between consecutive drinking tasks belonging to the same trial, a resting period of 4 s was observed. A single trial consisted of a sequence of a 30 s static acquisition, a synchronization movement, another 30 s static acquisition, and 25 consecutive repetitions of the drinking task. Considering the sub-phases as separated movements, 100 movements were expected within each trial. Each volunteer completed three consecutive trials with a 2-minute rest in between. Thus, a total of 75 trials (25 subjects × 3 trials) were performed.

### 2.4. Data Pre-Processing and Synchronization

The position of the IMU-mounted markers was reconstructed using the Vicon Nexus 2.12 software. The marker trajectories were visually checked and possible gaps were manually filled. To remove random noise, marker trajectories were low-pass filtered using a second-order zero-lag Butterworth filter with a cut-off frequency of 6 Hz. A Marker-cluster Local Frame (MLF) was defined using the markers attached to the IMU to determine its reference orientation with respect to the SP global reference frame. The orientation of the MLF was estimated using the singular value decomposition (SVD) [25]. Then, the angular velocities were obtained from the orientation data [26]. The angular velocities estimated from the SP system were cross-correlated with the angular velocities recorded through the IMU during the synchronization movement to temporally align the data between the acquisition systems.

To reduce rapid signal fluctuations that could lead to inaccurate movement segmentation, marker trajectories and IMU recordings were further smoothed by means of a fourth-order zero-lag low-pass Butterworth filter with a cut-off frequency of 1.5 Hz.

### 2.5. Movement Segmentation

#### 2.5.1. Stereophotogrammetric-Based Segmentation

Considering the SP system, voluntary movements were identified based on linear rather than angular velocity. Tri-axial linear velocity and its norm (|v|) were calculated from the marker trajectories. The linear velocity norm was normalized in amplitude between 0 and 1 considering the absolute maximum value $\left(\left|v_{max}\right|\right)$ observed during each trial. An adaptive threshold was then implemented to determine voluntary movement onset and offset. The threshold was defined as the average of optimal thresholds computed through Otsu’s method from the normalized linear velocity norm of each trial [27,28]. Otsu’s method, an unsupervised threshold selection method originally developed for image processing, uses the image gray-level histogram to distinguish objects from the background. In our case, Otsu’s method was applied to the normalized linear velocity norm to identify the threshold that best distinguishes between the “background” (i.e., involuntary movements or background noise) and the “main object” (i.e., voluntary movements). The optimal threshold was selected as the one maximizing the inter-class variance. In this study, the adaptive threshold for the SP-based segmentation ($Th_{SP}$) was set equal to 0.11·|v_max|.

Additionally, the results of the SP-based segmentation were manually checked by an expert operator, using the videos of the acquisitions as a reference. These segmentations were considered as the gold standard (GS) for the IMU-based segmentation approaches.

Figure 2a shows an example of voluntary movement segmentation obtained considering the SP-derived linear velocity norm.

#### 2.5.2. IMU-Based Segmentation

The performance of the Dynamic Adaptive Movement Segmentation (DynAMoS) method was compared against two of the most widely used approaches: the fixed thresholding approach proposed by Schwartz et al. (M1) [13] and the adaptive thresholding approach proposed by Carpinella et al. (M2) [14]. All the tested methods relied on applying a threshold to the angular velocity norm (Ω), derived from the filtered angular velocity signals. In the following, the three tested segmentation approaches are described:

*(a) Fixed thresholding by Schwarz et al. (M1):* This method consists of the application of a single threshold ($Th_{M1}$) whose value was empirically set equal to 0.1 rad/s. Ω values higher than the threshold are identified as voluntary movements. The threshold value represents a reasonable value for discriminating between stationary and non-stationary states. Figure 2b represents the segmentation results obtained by applying the M1 method to the angular velocity norm of a representative healthy subject during a drinking task.

*(b) Adaptive thresholding by Carpinella et al. (M2):* This method consists of the application of an adaptive threshold defined as $Th_{M2}=0.25\cdot \Omega_{max}$, where $\Omega_{max}$ represents the maximum value of the angular velocity norm recorded over the task’s duration. Setting the threshold to 25% of the maximum value of the angular velocity norm represents a more conservative approach to the segmentation task, ensuring that all the segmented sections are actual movements. Figure 2c represents the segmentation results obtained by applying the M2 method to the angular velocity norm of a representative healthy subject during a drinking task.

*(c) Dynamic Adaptive Movement Segmentation (DynAMoS):* This newly proposed algorithm is based on an adaptive threshold and post-processing based on statistics applied to the duration of the identified movements to compensate for erroneous segmentations. The post-processing step implemented in DynAMoS was originally developed for clinical gait analysis to improve gait cycle segmentation [29]. The following outlines the detailed steps of DynAMoS:

Movement onset and offset are identified through the application of an adaptive threshold to the angular velocity norm (Ω). The adaptive threshold was defined as $Th_{DynAMoS}=k\;\Omega_{max}$, where k and $\Omega_{max}$ represent the multiplicative constant and the maximum value of the angular velocity norm recorded during each trial, respectively. k value was determined by applying Otsu’s method [27,28] to the angular velocity norm, as detailed in Section 2.5.1: Stereophotogrammetric-Based Segmentation. The optimal value selected was 0.11;The duration of each segmented movement (T) is calculated as the difference between the corresponding offset and onset time instants;Considering the movement durations, iterative statistics-based post-processing is applied to minimize erroneous segmentations. For each subject, a movement duration histogram is generated (see Figure 3) and the median duration (M) is calculated;To identify outliers in the movement duration distribution, two thresholds are defined based on the median duration M: the lower and the upper threshold. The lower threshold is defined as αM (with 0<α<1), while the upper threshold is defined as βM (with 1<β<2). The multiplicative constants α and β are optimized for the task under analysis;Movements characterized by “atypical” durations (i.e., those falling below the lower threshold or above the upper threshold) are processed iteratively. If a movement has a duration shorter than the lower threshold, the algorithm tries to merge it with the nearest movement. Merging is performed only if the new movement duration (i.e., the sum of the two previous movement durations) falls between the αM and βM thresholds. Otherwise, no merging occurs. If a movement has a duration longer than the upper threshold, the algorithm tries to split it into two shorter movements by using local minima in the angular velocity norm as potential splitting points, if these minima are available. Splitting is performed only if both the new movement durations fall between the αM and βM thresholds. Otherwise, no splitting occurs;After each splitting or merging event, M, αM, and βM values are updated;The algorithm runs iteratively until all the movement duration outliers are processed.

Further details about DynAMoS functioning are freely available on the BIOLAB GitHub repository (https://github.com/Biolab-PoliTO/DynAMoS (accessed on 9 August 2024)).

Figure 2d represents an example of segmentation obtained using the previously described method.

The optimization of the parameters α and β was performed using a grid search approach on all the IMU data. Specifically, the value of α was chosen between 0.5 and 0.95 with steps of 0.05. Similarly, β was chosen between 1.05 and 1.5 with steps of 0.05. To find the best pair of parameters, the following cost function ($F_{c}$) was defined as detailed in Equation (1):(1)$$F_{c}=\frac{{Extra}_{IMU}}{{Total}_{IMU}}+\frac{{Missing}_{IMU}}{{Total}_{IMU}}+\bar{∆Onset}+\bar{∆Offset}+\bar{∆Duration}$$
where ${Extra}_{IMU}$ represents the number of “extra” movements obtained from the IMU data compared to the number of movements identified by the GS over all the trials. ${Missing}_{IMU}$ represents the number of missing movements compared to the number of movements identified by the GS over all the trials. ${Total}_{IMU}$ is the total number of movements obtained from the IMU data. $\bar{∆Onset}$ and $\bar{∆Offset}$ represent the onset and offset mean errors (expressed in seconds) between the movements obtained from IMU data and the GS, respectively. $\bar{∆Duration}$ represents the mean duration difference (expressed in seconds) between the movements obtained from IMU data and the GS. The best pair of parameters α and β was determined by finding the minimum value of the cost function $F_{c}$.

### 2.6. Performance Assessment

To evaluate the performance of each of the presented methods, several parameters were used. For each trial, it was calculated the number of movements the specific method detected ($N_{mov}$) compared to the GS, the percentage of erroneous movements (${Err}_{Mov}$) with respect to the GS, the onset and offset mean absolute error (${MEA}_{Onset}$ and ${MEA}_{Offset}$, respectively) against the GS. Notice that ${MEA}_{Onset}$ and ${MEA}_{Offset}$ were calculated only for those movements that were consistent between the GS- and the IMU-based segmentation. The duration of each movement (T) was calculated as the difference between the offset and onset time instants. Additionally, the execution time for segmenting a single movement trial was measured using an Intel^®^ Core™ i7-13700H CPU @ 2.40 GHz (Intel ©, Santa Clara, CA, USA). In the following analyses, the average over the trial of the temporal parameters (i.e., percentage of erroneous movements, onset/offset mean absolute error, movement duration, and execution time) was considered.

### 2.7. Drinking Task Characterization

After the evaluation of the effectiveness of the identification of voluntary movements from the inertial data, the drinking task was characterized in terms of the duration of each sub-phase. First, each repetition of the drinking task was split using the longer resting time (approximately 4 s) between consecutive movements. Then, the single sub-phases were identified and classified using the SP as a reference. For each segmentation method, the mean duration of each sub-phase over trials was calculated as the difference between the offset and onset time instants and compared against the GS.

### 2.8. Statistical Analysis

We applied the Kolmogorov–Smirnov test to assess the data distribution normality of the number of movements, the percentage of erroneous movements, the onset and offset mean absolute error, the movement duration, and the execution time. Based on the Kolmogorov–Smirnov test results, a *one-way* ANOVA (in case of normal distributions) or a Kruskal–Wallis test (for non-normal distributions) was used followed by post-hoc analysis with Bonferroni adjustments for multiple comparisons. All the analyses were performed setting the significance level (α) at 0.05. Parameter estimates were represented as mean ± standard error over the population. The effect size of the statistically significant differences was calculated through the Hedges’ g statistic [30]. A g value of 0.2, 0.5, and 0.8 were considered a small, medium, and large effect size, respectively.

The statistical analyses were performed using the Statistical and Machine Learning Toolbox of MATLAB release r2023b (The MathWorks Inc., Natick, MA, USA).

## 3. Results and Discussion

### 3.1. DynAMoS Optimization Process

Figure 4 shows the results of the DynAMoS optimization process aimed at selecting the optimal α and β values. The optimal parameters selected were α=0.8 and β=1.4. All DynAMoS results described in the following sections were obtained using these optimal parameter values.

### 3.2. Performance Assessment

The performance of IMU-based segmentation approaches against the stereophotogrammetric system (i.e., gold standard) is reported in Table 1 with the indication of the statistically significant differences between methods as tested through the Kruskal–Wallis test. Table 2 presents the results of the Bonferroni post-hoc test, indicating significant differences in segmentation performance between methods, represented by symbols.

The SP-based segmentation identified on average 100 ± 4 voluntary movements (mean ± standard error) per trial. The IMU-based segmentation approaches identified 168 ± 24 movements, 101 ± 7 movements, and 103 ± 5 movements considering M1, M2, and DynAMoS methods, respectively. Overall, the newly presented algorithm identified only 3% of erroneous movements (i.e., 2.8% extra movements and 0.2% missed movements) compared to the GS. M1 resulted in 39.8% of erroneous movements (i.e., 39.8% extra movements) compared to the GS, while M2 gave significantly better results, revealing only 3.7% of erroneous movements (i.e., 2.5% extra movements and 1.2% missed movements). As shown in Figure 5, in some cases the percentage of missed and extra movements per trial was relatively high, with a maximum of 51.5% and 15.1% of the movements, respectively. Statistically significant differences in the percentage of erroneous movements were detected among all the tested approaches (*p* < 0.0001). In particular, the worst performance was obtained considering the M1 approach. Even if no statistically significant differences in the percentage of erroneous movements were detected between M2 and DynAMoS, the difference in the number of outliers is noticeable (i.e., DynAMoS revealed a reduced number of outliers compared to M2).

These results confirm that the application of a low threshold, as employed by M1, can result in the detection of an excessive number of voluntary movements, which may be attributed to noise or involuntary movements. In contrast, the detection method M2 is characterized by a high number of missing movements, despite the adaptive threshold.

Considering the onset mean absolute errors, statistically significant differences were detected between all the tested approaches (*p* < 0.0001). Post-hoc analysis identified significant differences between DynAMoS and M1 (*p* < 0.0001; g = 3.9), between DynAMoS and M2 (*p* < 0.0001; g = 1.3), and between M1 and M2 (*p* < 0.0001; g = 2.5). Figure 6a represents the ${MAE}_{Onset}$ distributions with the indication of the statistically significant differences. Similarly, when considering the offset mean absolute error, significant differences were detected between DynAMoS and all the tested approaches (*p* < 0.0001). Bonferroni adjustments for multiple comparisons revealed significant differences between DynAMoS and M1 (*p* < 0.0001; g = 3.7), between DynAMoS and M2 (*p* < 0.0001; g = 3.7), and between M1 and M2 (*p* < 0.0001; g = 1.4). Figure 6b represents the ${MAE}_{Offset}$ distributions with the indication of the statistically significant differences. The distributions of the onset and offset mean absolute errors obtained using DynAMoS are largely concentrated below 0.1 s, a value that deviates considerably from those obtained considering M1 (mean absolute error higher than 0.2 s). In contrast, M2 errors have a different behavior. The mean onset error is 0.1 s, with most of the distribution lower than 0.15 s, whereas the mean offset error is 0.2 s.

Considering the movement durations, the tested approaches showed statistically significant differences (*p* < 0.0001). In particular, statistically significant differences were detected between M2 and the GS (*p* < 0.0001; g = 3.6), between M2 and M1 (*p* < 0.0001; g = 2.3), and between M2 and DynAMoS (*p* < 0.0001; g = 3.3). The difference in duration of the movement between M2 and the GS was approximately 30% of the mean duration of the movement identified on the GS. Although there was no statistically significant difference in the mean durations, the distribution obtained when applying M1 was more variable than the one obtained considering the GS and DynAMoS. A lower and fixed threshold results in longer movements and faster transitions given by small fluctuations around the threshold level, increasing the variability in the results. Figure 7 shows the distributions of the movement duration for all the tested approaches, with the indication of the statistically significant differences.

### 3.3. Drinking Task Characterization

Figure 8 and Table 3 report the results of the drinking task characterization for all the tested methods with the indication of the statistical significance, as identified by the Kruskal–Wallis test. Table 4 presents the results of the Bonferroni post-hoc test, indicating significant differences in movement sub-phase durations between methods, represented by symbols.

For all the sub-phases, a statistically significant difference in sub-phase duration was observed. Multiple comparisons resulted in a significant difference between M1 and M2, and between both M1 and M2 and the GS and DynAMoS (*p* < 0.0001; g > 1.7). In particular, the data distribution obtained considering M2 is below the lower quartile of the other distributions, while the data distribution obtained considering M1 is above the upper quartile of the other distributions. On the other hand, for all the sub-phases, there was no statistically significant difference (*p* > 0.8) between the timings obtained through the GS and those obtained through DynAMoS. The application of M2 results in a difference in the duration of the sub-phases that deviates from that obtained through the GS of approximately 20%, for Phase II and III, and 40%, for Phase I and IV. It is worth noting that similar sub-phases, similar in terms of the range of motion but opposed in terms of the goal of the movement, present similar timings and errors. In contrast, the result obtained by applying M1 overestimates movement sub-phase durations by approximately 45%. The estimates obtained by means of DynAMoS align with the GS results.

## 4. Final Considerations

This study presents a novel segmentation method developed to overcome the limitations of the most popular existing methods. In particular, the presented algorithm was compared with the threshold-based segmentation approaches proposed by Schwarz et al. [13] (M1) and Carpinella et al. [14] (M2).

Although the mean number of movements identified by M2 was the closest to that obtained by the GS, it is worth observing, from the ${Err}_{Mov}$ distribution represented in Figure 5, that in numerous instances the number of missing movements was significant. While selecting an adaptive threshold may be more appropriate for different movements, a high threshold can result in a higher percentage of missed movements (51% in the case of the M2 approach), especially when the movement consists of different sub-phases with varying velocities. In contrast, results obtained through the M1 approach indicate a consistent over-detection of voluntary movements, revealing that the threshold is too low and likely influenced by small signal fluctuations close to the threshold (see Figure 2). As can be seen observing Figure 5, DynAMoS ${Err}_{Mov}$ distribution is similar in variability to the results obtained with M2. However, the number of outliers and the percentage of missed movements are considerably reduced, revealing the reliability of the segmentation.

Focusing on movement onset and offset detection, the method presented in this study was more accurate than the other tested approaches when compared to the GS. In fact, the movement durations of the four sub-phases obtained through DynAMoS were closer to the GS (with an average difference of 0.04 s). Research studies previously published show that a 15% variation in the movement performance metrics is considered a clinically important difference [12]. In our study, movement durations obtained by means of methods M1 and M2 differ from the GS from a minimum of 20% to over 40%, whereas a maximum of 3% variation was obtained considering DynAMoS. Therefore, the difference in movement timing between the state-of-the-art methods we considered and the GS is considerably higher than 15%, suggesting that the use of these segmentation methods may strongly impact the clinical assessment. In contrast, the small differences in movement timings between DynAMoS and the GS make the method herein proposed considerably more reliable and potentially applicable in the clinical assessment of patients.

Even though a clinical validation of the method was not performed, it is possible to compare the obtained results with similar results presented in the literature. For example, Patterson et al. [31] evaluated post-stroke patients and healthy controls reaching a target at a comfortable speed by using an SP system. On average, the durations of the movements were 0.96 ± 0.27 s and 0.67 ± 0.12 s considering the post-stroke patients and the healthy controls, respectively. The difference in movement duration between the stroke survivors and the healthy controls is lower than the difference in durations observed in our results. Thus, the application of different segmentation approaches may not be able to differentiate between a healthy and a pathological population. Furthermore, Carpinella et al. [14] demonstrated a statistically significant difference of approximately 0.4 s in grasp movement duration between multiple sclerosis patients and healthy controls. This difference is not substantially larger than the sub-phase duration error between M1, M2, and the GS.

It is worth noting that, since the two state-of-the-art approaches were initially developed using IMU data from movements similar to those in our study (i.e., reaching and manipulation tasks) and with comparable velocities, we deemed it reasonable to apply the threshold values proposed in their respective papers without additional optimization.

Although the results are promising, there are some limitations associated with the method. The first is the impossibility of applying the algorithm in real time, due to the adaptive thresholding and the post-processing step. In fact, both these steps require the whole inertial data to compute the required parameters (i.e., the maximum of the angular velocity norm and the movement duration distribution). Also, DynAMoS was developed for the segmentation of movements commonly analyzed in functional rehabilitation protocols that are typically executed by patients several times with limited variability in duration. The analysis of ballistic or highly variable movements should be tested. Additionally, this study was conducted on healthy subjects only. Accurate movement segmentation is crucial for effective movement characterization in the assessment of patients. Thus, validating the effectiveness of DynAMoS on patients represents a critical step toward developing a clinically valuable tool for telerehabilitation protocols. Further studies are needed to clinically validate this approach for patient assessment.

## 5. Conclusions

In this study, we compare a new IMU-based segmentation method for upper-limb movements with two popular segmentation methods [13,14]. The movement herein considered is the reach-to-grasp movement, due to its frequent use in the clinical evaluation of patients suffering from upper-limb motion disorders. The results show that the proposed method performs significantly better than the two implemented ones. According to Kwakkel et al. [12], a segmentation accuracy similar to that of DynAMoS could make it available for clinical applications. Using IMU for motion detection and the proposed algorithm for time segmentation of upper-limb voluntary movements could make home monitoring applications more effective for assessing the motion improvements of patients following domicile rehabilitation protocols.

## Figures and Tables

**Figure 1 sensors-24-06119-f001:**
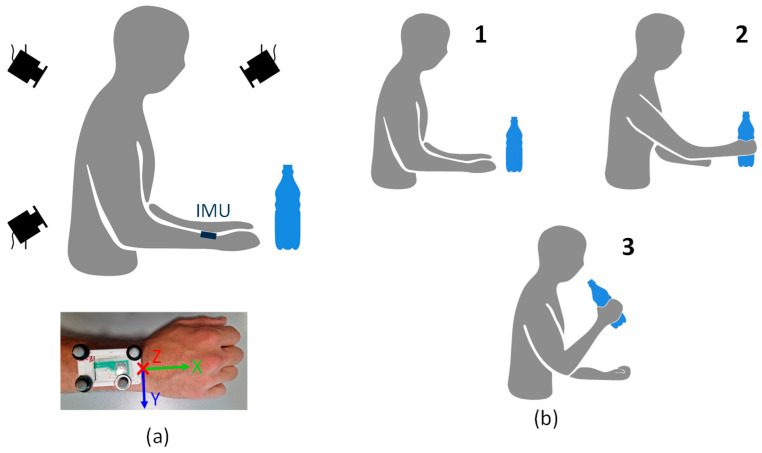
Panel (**a**): Representation of the acquisition setup (composed by the SP system and the IMU) and a picture of the wrist-worn IMU with 4 photo-reflective markers. Panel (**b**): Representation of the drinking task sub-phases. The drinking task consisted of reaching for and grasping the bottle (step 1 and 2), lifting the bottle simulating drinking (step 2 and 3), placing the bottle back on the table (step 3 and 2), and returning to the resting position (step 2 and 1).

**Figure 2 sensors-24-06119-f002:**
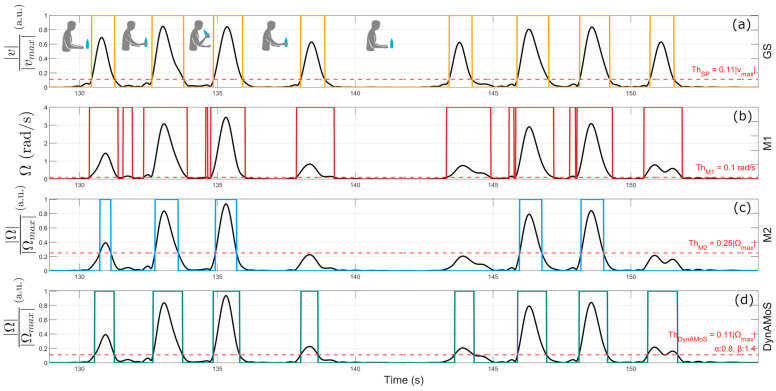
Example of segmentation results obtained through the application of the (**a**) gold standard and (**b**–**d**) three tested methods (i.e., the method by Schwarz et al.-M1, the method by Carpinella et al.-M2, and the newly proposed method––DynAMoS, respectively) on inertial data acquired during drinking tasks. Dotted horizontal lines represent the threshold values of each method, while colored binary masks represent the segmentation output of each method.

**Figure 3 sensors-24-06119-f003:**
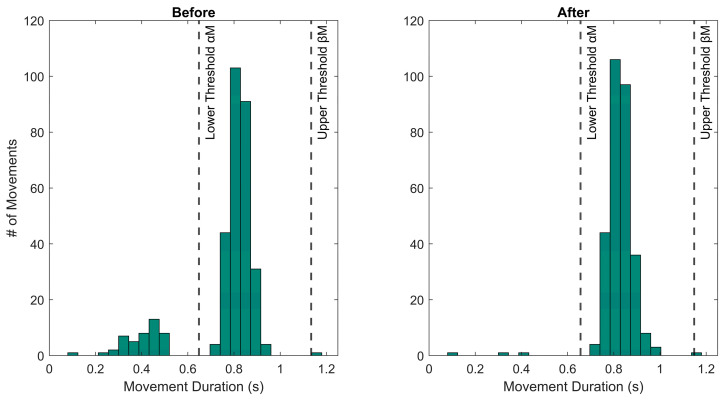
Example of movement duration histograms before (**left side**) and after (**right side**) the application of the statistics-based post-processing.

**Figure 4 sensors-24-06119-f004:**
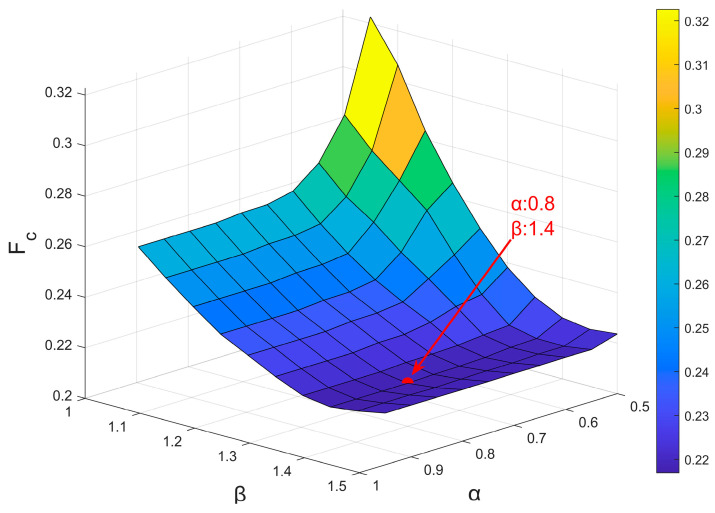
Values assumed by the cost function $F_{c}$ when considering different α and β. The red dot identifies the minimum of the cost function $F_{c}$ found when α = 0.8 and β = 1.4.

**Figure 5 sensors-24-06119-f005:**
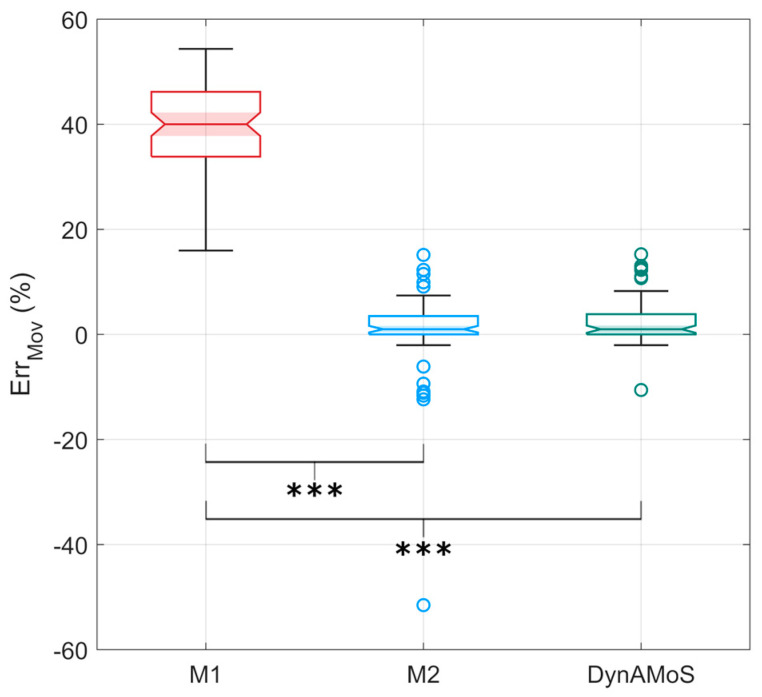
Boxplots representing the percentage of erroneous movements computed between each tested segmentation approach (M1, M2, and DynAMoS) and the gold standard. Statistically significant differences are represented through asterisks (*** *p* < 0.0001). The average execution time computed from the three IMU-based segmentation methods was 1.1 ± 0.6 ms, 1.0 ± 0.6 ms, and 2.7 ± 2.1 ms for M1, M2, and DynAMoS, respectively. Statistically significant differences were detected between M1 and DynAMoS (*p* < 0.0001; g = 1.4) and between M2 and DynAMoS (*p* < 0.0001; g = 1.5).

**Figure 6 sensors-24-06119-f006:**
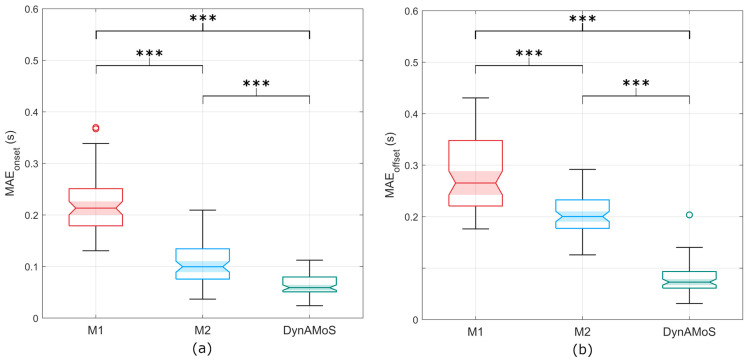
Boxplots representing (**a**) the ${MAE}_{Onset}$ and (**b**) the ${MAE}_{Offset}$ computed between each tested segmentation approach (M1, M2, and DynAMoS) and the gold standard. Statistically significant differences are represented through asterisks (*** *p* < 0.0001).

**Figure 7 sensors-24-06119-f007:**
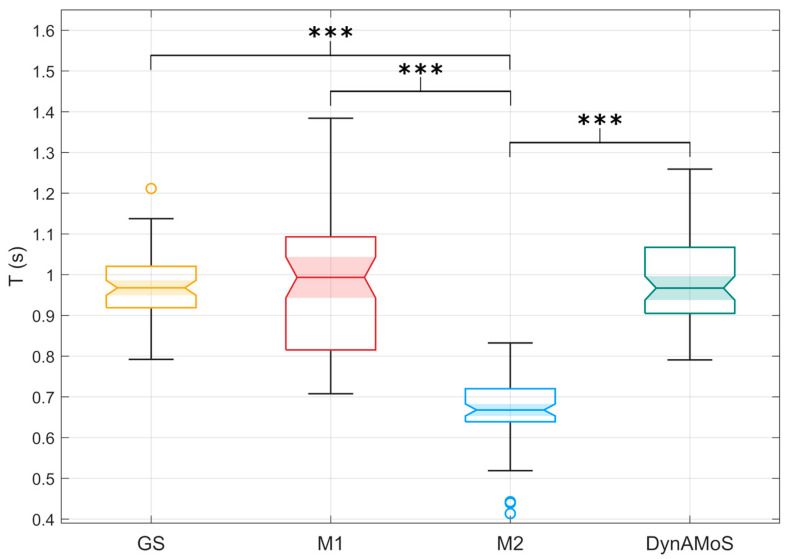
Boxplots representing the movement durations (T) computed considering the GS (yellow), M1 (red), M2 (blue), and DynAMoS (green) methods. Statistically significant differences are represented through asterisks (*** *p* < 0.0001).

**Figure 8 sensors-24-06119-f008:**
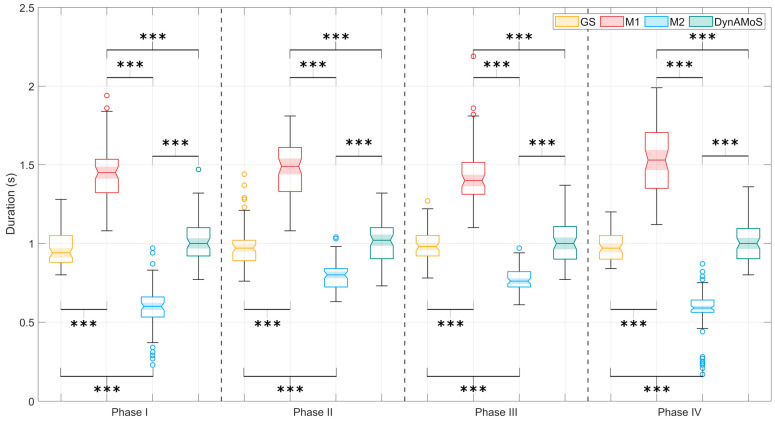
Distribution of the mean duration of the sub-phases of the drinking task for the sample population for the tested methods and the SP gold standard. Statistically significant differences are represented through asterisks (*** *p* < 0.0001).

**Table 1 sensors-24-06119-t001:** Performance assessment of the three tested approaches against the stereophotogrammetric system.

Performance Assessment	Segmentation Method				Kruskal–Wallis
	GS	M1	M2	DynAMoS	*p*-Value
$N_{mov}$	100 ± 4	168 ± 24	101 ± 7	103 ± 5	**<0.0001**
${Err}_{Mov}$ (%)	N/A	39.8	3.7	3.0	**<0.0001**
${MAE}_{Onset}$ (s)	N/A	0.22 ± 0.05	0.10 ± 0.04	0.07 ± 0.02	**<0.0001**
${MAE}_{Offset}$ (s)	N/A	0.29 ± 0.07	0.20 ± 0.04	0.08 ± 0.03	**<0.0001**
T (s)	0.97 ± 0.08	0.98 ± 0.17	0.67 ± 0.08	0.98 ± 0.11	**<0.0001**

Parameters are represented as mean ± standard error over the population. N/A: not applicable; GS: gold standard.

**Table 2 sensors-24-06119-t002:** Bonferroni post-hoc analysis results for performance metrics.

Segmentation Methods	GS	M1	M2	DynAMoS
GS	N/A	□	□, •	□
M1	□	N/A	□, ◊, †, ‡, •	□, ◊, †, ‡
M2	□, •	□, ◊, †, ‡, •	N/A	†, ‡, •
DynAMoS	□	□, ◊, †, ‡	†, ‡, •	N/A

N/A: Not applicable; GS: Gold standard; M1: Fixed thresholding method by Schwarz et al.; M2: Adaptive thresholding by Carpinella et al. Squares (□), diamonds (◊), daggers (†), double daggers (‡), and points (•) represent statistically significant differences between pairs of methods based on $N_{mov}$, ${Err}_{Mov}$, ${MEA}_{Onset}$, ${MEA}_{Offset}$, and T, respectively.

**Table 3 sensors-24-06119-t003:** Drinking task sub-phase durations.

Sub-phase Duration (s)	Segmentation Method				Kruskal–Wallis
	GS	M1	M2	DynAMoS	*p*-Value
Phase I	0.97 ± 0.11	1.44 ± 0.19	0.59 ± 0.14	1.01 ± 0.14	**<0.0001**
Phase II	0.98 ± 0.13	1.47 ± 0.17	0.79 ± 0.08	1.02 ± 0.13	**<0.0001**
Phase III	0.98 ± 0.10	1.44 ± 0.20	0.77 ± 0.07	1.01 ± 0.13	**<0.0001**
Phase IV	0.98 ± 0.09	1.54 ± 0.22	0.58 ± 0.14	1.00 ± 0.13	**<0.0001**

Parameters are represented as mean ± standard error over the population. GS: Gold standard; M1: Fixed thresholding method by Schwarz et al.; M2: Adaptive thresholding by Carpinella et al.

**Table 4 sensors-24-06119-t004:** Bonferroni post-hoc analysis results for movement sub-phase duration.

Segmentation Methods	GS	M1	M2	DynAMoS
GS	N/A	□, ◊, †, ‡	□, ◊, †, ‡	
M1	□, ◊, †, ‡	N/A	□, ◊, †, ‡	□, ◊, †, ‡
M2	□, ◊, †, ‡	□, ◊, †, ‡	N/A	□, ◊, †, ‡
DynAMoS		□, ◊, †, ‡	□, ◊, †, ‡	N/A

N/A: Not applicable; GS: Gold standard; M1: Fixed thresholding method by Schwarz et al.; M2: Adaptive thresholding by Carpinella et al. Squares (□), diamonds (◊), daggers (†), and double daggers (‡) represent statistically significant differences between pairs of methods based on Phase I, Phase II, Phase III, and Phase IV, respectively.

## Data Availability

DynAMoS algorithm, detailed documentation, and a sample dataset are freely available on the BIOLAB GitHub repository (https://github.com/Biolab-PoliTO/DynAMoS (accessed on 9 August 2024)).

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
