# Peer review of "A Statistical Approach for Functional Reach-to-Grasp Segmentation Using a Single Inertial Measurement Unit"

_sensors, 2024, doi:10.3390/s24186119_

Round 1

Reviewer 1 Report

Comments and Suggestions for Authors

The article proposed a statistical approach for functional reach-to-grasp segmentation using a single inertial measurement unit. This article can be considered for publication in "sensors" after revising the following questions.

1) Although some advances in current motion analysis systems are mentioned in the introduction section, the literature review section could be more comprehensive, especially for studies using inertial measurement units (IMUs) for upper limb motion analysis. We suggest adding more recent research results in related fields, especially the latest advances in motion segmentation algorithms (especially adaptive threshold methods and statistical methods) to reflect the innovation and necessity of this study.

2) Describe each step in the DynAMoS algorithm in detail, including how thresholds are selected, how statistics are calculated, and how conditions for merging and splitting moving segments are determined. 

3) In addition to the evaluation indicators, it is recommended to add other evaluation criteria, such as time complexity, memory consumption, sensitivity to noise, etc., to comprehensively evaluate the performance of DynAMoS algorithm. At the same time, a more detailed comparison can be made with other advanced algorithms to highlight the advantages and limitations of DynAMoS.

4) Although the effectiveness of the DynAMoS method was validated in healthy subjects, clinical validation was lacking. The authors should discuss the potential application of this method in patient evaluation and recommend clinical validation in future studies to evaluate its performance in different pathological conditions.

5) The language of the full text should be more accurate and concise. Avoid using vague or ambiguous expressions and make sure that each sentence clearly conveys the author's meaning. In addition, pay attention to checking grammar and spelling errors to improve the overall quality of the manuscript.

Comments on the Quality of English Language

The language of the full text should be more accurate and concise. Avoid using vague or ambiguous expressions and make sure that each sentence clearly conveys the author's meaning. In addition, pay attention to checking grammar and spelling errors to improve the overall quality of the manuscript.

Reviewer 2 Report

Comments and Suggestions for Authors

Below you can find some remarks for your Submission:

1.                Do you use any filtration of the IMUs output signals?

2.                The known disadvantage of MEMS IMUs is non-stable parameters (Biases and Scale Factors) from SWITCH-OFF to SWITCH-ON. Do you calibrate your IMU before processing?

Reviewer 3 Report

Comments and Suggestions for Authors

This study proposes a statistical segmentation method for angular velocity signals during reach-to-grasp motions. The authors compared the proposed method with several existing methods including optical motion tracking methods as a golden standard. The results show superior performance than existing methods and are comparable to the golden standard.

There are some comments and questions as follows.

(1) The proposed method utilizes a dynamic adaptive segmentation threshold, based on statistical distribution of the movement duration. It seems that this approach assumes and is limited to that the target movements have the same or similar duration. However, movements have different durations in general, and this assumption seems to strictly limit the applicability of the proposed method in general cases.

(2) Regarding the experimental protocol, were the durations of the four-movement phases tried to keep constant?

(3) Fixed segment thresholds 0.1 rad/s and adaptive threshold with fixed ratio 0.25 x (angular velocity norm) were used for methods 1 and 2, respectively. However, it seems reasonable that the fixed threshold and the ratio should have been optimized even with some simple method for fair comparison.

(4) How many samples were used for the number of detected movements and its error? 5 samples from 5 subjects were used since each subject tried 100 movements. If this is true, the 5 samples are too few for comparison.

(5) In Tables 1 and 2, the meaning of the significance notations such as asterisks and double asterisks is not clear.

(6) Figures 7 and 8, and Table 2 show the duration of movements. However, since the durations are not constant in this experiment, comparisons of the durations seem not proper. The differences in durations seem to be more meaningful.

Sincerely,

The reviewer.

Round 2

Reviewer 3 Report

Comments and Suggestions for Authors

The authors have cleared all the comments and questions from this reviewer except one.

The representations of the significant differences on the table 1 and 2 are still not clear enough. From the description in the main text and caption in the tables, it is not easy to interpret the symbols. For instance, the square symbols indicate a significant difference between the GS and M1 methods. It would be helpful to provide information such as the squares representing statistically significant differences between methods (GS vs M1). Otherwise, a table or matrix diagram of which rows and columns have the four methods, and which contents have the symbols, would be helpful.

Sincerely,

The reviewer.
